# The Radiative and Geometric Properties of Melting First-Year Landfast Sea Ice in the Arctic

Nathan J. M. Laxague<sup>1,2</sup>, Christopher J. Zappa<sup>1</sup>, Andrew R. Mahoney<sup>3</sup>, John Goodwin<sup>4</sup>, Cyrus Harris<sup>4</sup>, Robert E. Schaeffer<sup>4</sup>, Roswell Schaeffer, Sr.<sup>4</sup>, Sarah Betcher<sup>5</sup>, Donna D. W. Hauser<sup>6</sup>, Carson R. Witte<sup>1</sup>, Jessica M. Lindsay<sup>7</sup>, Ajit Subramaniam<sup>1</sup>, Kate E. Turner<sup>8</sup>, and Alex Whiting<sup>9</sup>

**Correspondence:** Nathan J. M. Laxague (Nathan.Laxague@unh.edu)

Abstract. In polar regions, sea ice is a crucial mediator of the interaction between earth's atmosphere and oceans. Its formation and breakup is intimately connected with large-scale climatic processes, local weather patterns, and the use of sea ice in coastal Arctic regions by Indigenous People. In order to investigate the physical phenomena at the heart of this process, a set of targeted, intensive observations were made over Spring sea ice melt and breakup in Kotzebue Sound, Alaska. These observations were planned and executed through a collaborative effort in which an Indigenous Elder advisory council from Kotzebue and scientists participated in co-production of hypotheses and observational research, including a stronger understanding of the physical properties of sea ice during spring melt. Here we present the results of observations performed using high-endurance, fixed-wing uncrewed aerial vehicles (UAVs) containing custom-built scientific payloads. Repeated flights over the measurement period captured the early stages of the transition from a white, snow-covered state to a broken up, bare/blue-green state. We found that the reflectance of sea ice features depend strongly on their size. Snow patches get darker as they get smaller, an effect owed to the geometric relationship between bright interior and the darker, melting feature edges. Conversely, bare patches get darker as they get larger. For the largest ice features observed, bare blue-green ice patches were found to be  $\approx 20\%$  less reflective than average across all observational cases, while large snowy/white ice patches were found to be  $\approx 20\%$  more reflective than that same average.

<sup>&</sup>lt;sup>1</sup>Lamont-Doherty Earth Observatory of Columbia University, Palisades, NY, USA

<sup>&</sup>lt;sup>2</sup>University of New Hampshire, Durham, NH, USA

<sup>&</sup>lt;sup>3</sup>Geophysical Institute, University of Alaska Fairbanks, Fairbanks, AK, USA

<sup>&</sup>lt;sup>4</sup>Community of Kotzebue, Kotzebue, AK, USA

<sup>&</sup>lt;sup>5</sup>Farthest North Films

<sup>&</sup>lt;sup>6</sup>International Arctic Research Center, University of Alaska Fairbanks, Fairbanks, AK, USA

<sup>&</sup>lt;sup>7</sup>University of Washington, Seattle, WA, USA

<sup>&</sup>lt;sup>8</sup>National Institute of Water and Atmospheric Research, Wellington, NZ

<sup>&</sup>lt;sup>9</sup>Native Village of Kotzebue, Kotzebue, AK, USA

#### 15 1 Introduction

45

### 1.1 Background

Sea ice is an essential feature of the polar oceans, looming large as both visual wonder and geophysical presence. The standard processes of air-sea interaction—whereby fluxes of momentum, heat, and mass are mediated by turbulent flow past an undulating interface—are fundamentally transformed by the presence of sea ice which attenuates (or completely suppresses) surface waves and wind-forced currents. However, there is perhaps no sea ice-related geophysical effect more stark than its alteration of the solar radiative balance. Whereas the ice-free ocean tends to reflect less than 10% of incoming solar radiation, the myriad forms of snow and ice at various stages of melt and degradation will reflect between 15-90% of incoming solar radiation. Broadband albedo, the ratio of wavelength-integrated reflected solar irradiance to incident solar irradiance, is strongly dependent on the surface state of the ice (Perovich et al., 1998), with fractional cover of snow on the surface the single most important determinant of high albedo (Yackel et al., 2000). The formation of melt ponds on the surface of sea ice alters surface radiative properties, lowering surface albedo (Barber and Yackel, 2010) and increasing the transmittance of light through the ice into the underlying sea (Ehn et al., 2011; Frey et al., 2011; Light et al., 2015).

It has long been known that solar radiative heat flux into the sea ice surface contributes to melting, reducing its reflectance and increasing its susceptibility to further degradation (Budyko, 1969). This phenomenon, known as the ice-albedo feedback, is understood to be a key component of the Earth's climatic variability. The rapid decline in sea ice extent in the recent past (Brennan et al., 2020) has led to an aggregate scale change in surface albedo as ice melts completely and is replaced by open water. Additionally, a widespread decline in Arctic sea ice thickness has been observed over the past half-century (Kwok and Rothrock, 2009). This is largely due to another feedback: as sea ice thins, it reflects less solar radiation (Lu et al., 2016) and allows for greater transmission of light (Light et al., 2008). This in turn results in increased ocean heat and enhanced melting of the sea ice bottom (Planck et al., 2020). Recent changes to the thickness, surface state, and overall extent of sea ice have yielded a dramatic increase in the absorption of solar heat in the Arctic Ocean (Perovich and Polashenski, 2012). This effect has contributed to the Arctic's warming at nearly four times the global average rate (Rantanen et al., 2022), a phenomenon with the name "Arctic Amplification". There is abundant evidence that this is due to the diminishment of sea ice cover and thickness (Dai et al., 2019; Perovich et al., 2020; Screen and Simmonds, 2010). All of these trends have led to Arctic ice cover becoming far less resilient to change than it was decades ago (Overland, 2020). The consequences of these effects are most concrete, immediate, and stark for Indigenous Arctic communities who exist among the sea ice and often rely on it as part of their way of life (Gearheard et al., 2013). For decades, the melt season has been trending to occur sooner and last for longer each year (Markus et al., 2009; Stroeve et al., 2014), threatening long-practiced Indigenous traditions such as seasonal hunts (Hauser et al., 2021).

In order to improve our understanding of the processes which drive these changes, there is a need for new field observations which quantify surface fluxes in high-latitude environments (Bourassa et al., 2013). So-called "process" studies which target particular physical phenomena are particularly illuminating (Carmack et al., 2015). The recent results of Horvat et al. (2020) demonstrate that the geometric properties of features on the sea ice surface have a net effect on radiative flux. Furthermore, the

geometry and abundance of those features may often be described via simple scaling frameworks (Popović et al., 2018). These results are key motivators of the present analysis. When conducting research on adaptation to climate change, it is valuable to move from an extractive (David-Chavez and Gavin, 2018) framework to one of co-production and stakeholder engagement (Klenk et al., 2017). The co-production of knowledge with local research partners is at its most effective when it is both iterative and interactive (Bremer et al., 2019), incorporating the needs of the community with an understanding of global-scale challenges (Eicken et al., 2021). This shift in mindset benefits the local communities most directly impacted by the processes being studied- but it also benefits the scientific research itself, with Indigenous Knowledge-holders providing key insights into local dynamics (Eicken, 2010), especially when equity between Indigenous People and scientists is a key component of the co-production framework (Yua et al., 2022).

# 1.2 The present study

**Figure 1.** Map of **a** Alaska, **b** Kotzebue Sound, and three consecutive satellite overpasses showing the change of sea ice color and extent during May of 2019: **c** 5/6/2019, **d** 5/9/2019, **e** 5/14/2019.

The present study arose as part of the Ikaagvik Sikukun project, a collaborative effort of knowledge co-production and observational research completed by an Indigenous Elder advisory council (study co-authors J.G., C.H., R.J.S., and R.S., Sr.)

from Kotzebue, AK (Figure 1) and an interdisciplinary team of scientists. This project was community-involved from the outset, with the initial stage of collaboration centered around the generation of driving research questions largely focused on understanding the physical and biological changes associated with sea ice loss in Kotzebue Sound (Hauser et al., 2021; Lindsay et al., 2023; Mahoney et al., 2021; Witte et al., 2021). In many coastal regions, the onset of sea ice breakup is associated with a seasonal hunting period; in Kotzebue Sound, the Qikiqtaġruŋmiut hunt of *ugruk* (bearded seal) coincides with the breakup of sea ice. The details of this connection—and of our co-production of knowledge approach—is described in more detail by Hauser et al. (2021). These factors were to be investigated using a variety of in situ, satellite, and uncrewed airborne observations. It is the latter set of high-resolution observations made from a series of flights throughout a period of advanced melt that form the basis of the present study.

Satellite images of Kotzebue Sound taken during our field observational campaign (Figure 1) show that the region's landfast sea ice appeared mostly blue-green by 5/9, with a major breakup event occurring between 5/13 and 5/14. Several important physical parameters are shown as stacked timeseries in Figure 2, with vertical lines indicating the timing of satellite overpasses and UAV flights (with the "RAD" and "VNIR" payloads, to be defined in section 2). These were obtained from land-based instrumentation: air temperature and wind velocity data from the meteorological station at Kotzebue Ralph Wien Airport (station PAOT); downwelling hemispheric radiative fluxes from a pair of Kipp & Zonen sensors (pyranometer and pyrgeometer) mounted on the rooftop of the U.S. Fish and Wildlife office. For the majority of our observational period, the sky was quite cloudy; this can be seen in the downwelling radiative flux time series of Figure 2, with high thermal irradiance and spottiness in solar irradiance (that is, departure from its characteristic bell-shaped diurnal behavior). The major landfast ice breakup event (indicated by the gray shaded region on Figure 2) was preceded by a rapid transition in the wind forcing, with onshore wind of 5 m/s giving way to offshore wind of 7 m/s.

Figure 2. Environmental conditions during field campaign: a wind speed, b wind direction (coming-from convention), c air temperature, d downwelling solar irradiance, and e downwelling thermal irradiance. Wind/air measurements were made at Kotzebue Ralph Wien Airport, while radiative measurements were made at the U.S. Fish & Wildlife Service bunkhouse. Vertical lines indicate times of satellite overpasses (dotted violet), UAV flights with "RAD" payload (solid orange), UAV flights with "VNIR" payload (solid teal); gray shaded region denotes approximate time of major sea ice breakup.

# 2 Methods

#### 2.1 UAV-based radiometry

We utilized uncrewed aerial vehicles (UAVs) equipped with specialized payloads for sensing downwelling (sky-leaving) and upwelling (surface-leaving) radiative fluxes in solar/shortwave and thermal/longwave bands (details in table 1). The details of the UAS deployed during our field operations are provided by Zappa et al. (2020); what follows here is a summary of the elements most relevant to the present study. Each UAV carried in its nosecone a "Base" payload (holding the core power

and data management systems) and one of a number of modular payloads which allowed for specialized sensor depoloyment. Two specialized payloads—RAD, for "radiometric" and VNIR, for "visible-near-infrared"—were deployed for observing the radiative properties of the sea ice; their core onboard sensors are listed in table 1, while their capabilities are qualitatively depicted in Figure 3. The RAD payload contained up and down-looking Hukseflux model IR-02 pyrgeometers to measure thermal irradiance and up and down-looking Hukseflux model SR-03 pyranometers to measure solar irradiance, all sampled at 1 Hz. The VNIR payload contained an upward-looking Ocean Optics model USB2000+ spectrometer to measure downwelling visible and near-infrared spectral irradiance and a downward-looking Headwall Micro-Hyperspec imaging spectrometer to measure upwelling visible and near-infrared spectral radiance, all sampled at 25 Hz. The Headwall imaging spectrometer is a "pushbroom" sensor, with each image "frame" corresponding to 1004 cross-track pixels (at 0.032° IFOV) and 1004 spectral wavelengths (at 1.85 nm spectral resolution); of the 1004 spectral measurements, 216 fell within the wavelength range of 400-800 nm used for the analysis here. Each UAV's Base payload held an onboard inertial navigation unit (INU) comprised of a GNSS (Global Navigation Satellite Systems) receiver and an inertial measurement unit (IMU); a complementary ground station allowed for differential GNSS post-processing. After each flight, all GNSS and IMU data were combined via tightly-coupled processing, providing an integrated TSPI (time space position information) solution with centimeter-scale (±1 cm horizontal, ±2 cm vertical) position accuracy and 0.01° attitude accuracy.

The vast majority of the sea ice in Kotzebue Sound at the time of our field observations was landfast (specifically, shorefast) (Figure 1), and our UAV missions were planned to cover this contiguous region (Figure 4). We set out to separate the features observed at the surface of the landfast ice into distinct categories (e.g., snow, bare ice, melt ponds) and measure spectral albedo over those regions (Grenfell and Perovich, 1984, 2004; Perovich et al., 2002). The relatively high altitude of our measurements rendered the hemispheric observations of the RAD payload too coarse to parse the radiative signatures of individual feature types. Given an altitude of 150 m, and the assumption that our hemispheric radiometers are ideal cosine collectors, we estimate that 90% of the signal originates from a region below the aircraft with diameter of 270 m. In order to complement these observations, VNIR payload performed measurements of sea ice spectral radiance, allowing for discernment of radiative characteristics at decimeter-scale spatial resolution.

The irradiance data obtained via sensors on the RAD payload were used to compute net solar irradiance, net thermal irradiance, and shortwave albedo (Figure 3). The downwelling spectral irradiance and upwelling spectral radiance data obtained via sensors on the VNIR payload were used to compute the spectral surface reflectance (Figure 3). The mean surface VNIR reflectance and broadband albedo observed along the track of each of the UAV flights are shown in Figure 5. Note that the surface reflectance measurements are localized in space (no more than 250 meters on either side of the flight track) while the albedo measurements were obtained via hemispheric sensors, integrating information from a far broader spatial region.

Although our extended flight durations ( $\approx$ 1 hour) limited our ability to fly payloads in short succession, great care was taken to ensure that ambient radiative conditions did not differ too greatly from one flight to the next: the average times of day for flight operations were 13:43:38, 15:51:16, & 18:30:08 AKDT (RAD flights) and 17:55:53, 12:37:45, & 12:55:05 AKDT (VNIR flights); during operation, local solar noon was approximately 14:47:35 AKDT. These flight times correspond to average solar *Reviewer \$1*: zenithelevation angles of 40.86°, 40.78°, and 30.62° for the RAD flights and 32.39°, 37.52°, and 38.78° for the VNIR

**Figure 3.** Depiction of fixed and aerial observational platforms, with arrows indicating radiative fluxes. Red dashed region on UAV indicates modular nosecone, with two red dashed boxes showing the measurement capabilities of two payloads: "RAD" payload, with upward/downward-looking pyranometers and pyrgeometers for characterizing upwelling and downwelling thermal and solar radiative fluxes; "VNIR" payload, with a downward-looking hyperspectral pushbroom imager and upward/downward-looking spectrometers for characterizing the spectral surface reflectance. Downwelling thermal and solar radiative fluxes were also measured from a rooftop station at the U.S. Fish and Wildlife Service bunkhouse. Wind speed, direction, and air temperature were measured from a meteorological station at Kotzebue Ralph Wien Airport.

flights (NOAA, ESRL solar calculator). As shown in the time series of Figure 2, the incoming irradiance did not vary greatly across the VNIR flights ( $E_S = 570\pm50~\rm W~m^{-2}$ ,  $E_T = 275\pm5~\rm W~m^{-2}$ ). The disparate tracks from one flight day to the next resulted from rapidly-changing surface conditions; for the first two sets of flights, RAD and VNIR payloads followed the same flight paths. However, by 5/15, the degradation of the sea ice surface advanced to the point that our visual observer could not maintain line of sight with the aircraft, resulting in different flight plans for the RAD and VNIR payloads on that day. Reviewer \$1:\text{Due} to the non-overlapping nature of the final set of flightplans on 5/15, data from that day's flights were excluded from the analysis described here. The flight tracks are shown in Figure 4 Reviewer \$1:\text{for illustrative purposes}.

| Quantity                     | Dimensions                                          | Symbol         | Sensor                                                      | Payload & Orientation | Spectral<br>Sensitivity    | Surface Spatial<br>Resolution     |
|------------------------------|-----------------------------------------------------|----------------|-------------------------------------------------------------|-----------------------|----------------------------|-----------------------------------|
| Thermal irradiance           | ${ m W~m^{-2}}$                                     | $E_T$          | Hukseflux IR02<br>Pyrgeometer                               | RAD,<br>up & down     | $4.5$ - $40~\mu\mathrm{m}$ | N/A                               |
| Solar irradiance             | ${ m W~m^{-2}}$                                     | $E_S$          | Hukseflux SR03 Pyrgeometer                                  | RAD,<br>up & down     | 285-3000 nm                | N/A                               |
| Solar<br>spectral irradiance | $\mathrm{W}~\mathrm{m}^{-2}~\mathrm{nm}^{-1}$       | $E_S(\lambda)$ | OceanOptics USB2000 Irradiance Spectrometer                 | VNIR, up              | 200-1100 nm                | N/A                               |
| Solar<br>spectral radiance   | W m <sup>-2</sup> nm <sup>-1</sup> sr <sup>-1</sup> | $L_S(\lambda)$ | Headwall Micro-HyperSpec VNIR A-Series Imaging Spectrometer | VNIR, down            | 400-1000 nm                | 57 cm @ 1000 m<br>flight altitude |

**Table 1.** Inventory of radiative measurements made via UAV, including the spectral sensitivity and surface spatial resolution.

# 2.2 Sea ice surface feature identification and processing

140

145

Surface reflectance data obtained from the VNIR payload were georeferenced according to the aircraft's attitude and position from the tightly-coupled TSPI solution. At a flight altitude of 1000 m, the total position error of each georeferenced radiance observation was estimated to be 

**Figure 4.** Flight tracks colored by data products of three days of operation with RAD and VNIR payloads (RAD only on 5/6/2019). Each flight day is confined to a column; each row consists of a single variable type (wavelength-averaged surface reflectance and broadband surface albedo, respectively). The green region indicates the Baldwin Peninsula, the white region indicates landfast ice, and the blue region indicates open water in Kotzebue Sound.

For this, we turned to the technique of König and Oppelt (2020), whereby the slope of the spectral reflectance in the near-infrared may be interpreted to infer the melt pond depth. The combined two-layer processing involves a check for melt ponds (which are then excluded from further partitioning) followed by a thresholding along the NIR:B value of 0.9. A demonstration of this processing is shown in Figure A2.

This scheme yields a three-level classification, allowing for statistical analysis to be performed on the geometric and radiative characteristics of ponded, bare, and snowy ice features ranging from decimeters to hundreds of meters in length scale. MATLAB's *regionprops* function was used for this processing, yielding the area and perimeter of each detected feature as well as the spectral surface reflectance at each point within each feature. Regions smaller than nine pixels (area 

**Figure 5.** Spectral albedo **a** and the ratio between the spectral albedo in the near-infrared (NIR) and blue, or NIR:B **b** for a variety of classifications of first-year sea ice, organized by broadband albedo. Data shown in **a** were taken from Perovich (2017) and used to compute quantity shown in **b**. Asterisk in panel **b** indicates a NIR:blue ratio of 0.9.

160 feature perimeter, one may compute the characteristic length scale  $D = \frac{4 \cdot area}{perimeter}$ ; for a circular feature, D reduces to the diameter.

# 3 Results

#### 3.1 Spatially-averaged properties

By combining the observations from consecutive RAD-VNIR flights, it is possible to relate the surface spectral reflectance (and color) to the net solar irradiance. This is shown in Figure 7, with mean visible \*Reviewer \$2: and near-infrared\* surface reflectance plotted against net solar irradiance in Figure 7a and the spectral surface reflectance binned by net solar irradiance in Figure 7b. Marker color in Figure 7a was generated from the reflectance in the red, green, and blue bands. Spectrally-flat surfaces (appearing white or gray) are associated with lower net solar irradiance; surfaces colored blue-green are associated with higher net solar irradiance. This is borne out in the binned spectra of Figure 7b, with higher net solar irradiance occurring for lower surface reflectance—but especially lower surface reflectance in the red and near-infared regimes.

Figure 6. a Color mosaic produced from the red, green, and blue channels of  $\rho_s$ . b Ratio of the NIR and blue channels of  $\rho_s$ , (NIR:B). c Binarized frame showing separation into "bare" and "snowy" portions.

# 3.2 Geometric properties of surface features

The bulk of section 2.2 was devoted to describing a framework for sea ice feature classification and geometric/radiative analysis. We now show the results of employing that framework. The relationship between area and perimeter offers more than a convenient way of rendering feature length scale— it provides us with information about the fractal dimensionality of the features on the sea ice surface. One such approach for determining fractal dimension (here given as fd rather than the literature-standard D in order to avoid confusion with our length scale D) is the aptly-named area/perimeter relationship (Klinkenberg, 1994):  $P \propto (\sqrt{area})^{fd}$ . In practical terms, fd is computed as half the slope of the relationship between  $\log(perimeter)$  and  $\log(area)$ ; this approach has been applied to melt pond data produced from observations (Hohenegger et al., 2012) and physical geometric modeling (Bowen et al., 2018; Horvat et al., 2020). A value of fd = 1 indicates that perimeter scales as the square root of the area (true for non-overlapping simple shapes). In mathematical terms, fd = 2 represents a shape-filling curve; in our practical application, fd approaching 2 indicates the presence of quasi-1D features that are one ground sample distance wide.

The size distribution and area/perimeter relationship of surface melt ponds are provided in Figure 8; panel **a** also includes the melt pond size distributions produced from the aerial field observations of Perovich et al. (2002) and Huang et al. (2016) and the geometric model of Popović et al. (2018), all represented in terms of characteristic length scale  $D = \sqrt{\frac{4}{\pi} area}$ . The area/perimeter relationship in panel b is represented with a cloud of gray points indicating individual measurements overlaid with a bivariate histogram that excludes the 'linear features' which bump up against the sensor spatial resolution. The melt

Figure 7. a Variation of spatially-averaged surface reflectance  $\rho_s$  [sr<sup>-1</sup>]<sup>Reviewer #2:</sup> \_\_averaged over 400-800 nm wavelength and partitioned into 30-second discrete segments of flight—with net solar irradiance. Marker color corresponds to the surface color determined by the red, green, and blue channels of  $\rho_s$ . b Spatially-averaged spectral surface reflectance  $\rho_s(\lambda)$  [sr<sup>-1</sup>nm<sup>-1</sup>], with color indicating the bin-averaged net solar irradiance.

pond area/perimeter relationship has been shown to reveal a sigmoid transition in fractal dimension from 1 to 2, with the inflection point tending to occur around a feature area of  $100 \text{ m}^2$  (Hohenegger et al., 2012). Our observations do not show this relationship, likely due to the relative paucity of melt pond observations during our field campaign: less than 0.5% of the imaged sea ice surface area contained ponded ice. Indeed, the mean fractal dimension (or, as in Horvat et al. (2020), "pond distribution dimension") of  $\approx 1.5$  appears to indicate that a large fraction of the detected melt ponds had a small area.

# 3.3 Dependence of reflectance on feature size

The remaining 99.5% of sea ice surface that was not ponded therefore fell into our two other categories: bare and snowy. The multipanel Figure 9 contains size probability density functions and area/perimeter relationships for bare and snowy ice features. In panels **a-b**, we show feature size distributions in terms of characteristic length scale *D*. There appear to be two regimes in these distributions, with the probability density of large bare or snowy features falling off more steeply with diameter than that of the smaller, darker features. This transition may be analogous to the transition found for melt pond size distributions (Hohenegger et al., 2012; Popović et al., 2018) that follows the sharp increase in fractal dimension as a result of increasing melt pond complexity and connectedness. The geometric bivariate distributions shown in panels **c-d** provide a clearer picture of this

**Figure 8.** a melt pond size probability density functions, represented in terms of the pond effective diameter *D*. The observational data of Perovich et al. (2002) and Huang et al. (2016) are shown alongside the physical model of Popović et al. (2018) and observational data from the present study. **b** melt pond area-perimeter plot, with individual measurements represented by gray dots and the overlying color indicating the value of the corresponding bivariate histogram. Three lines overlaid on the figure correspond to fractal dimensions of 1, 2, and 1.5, the "pond distribution dimension" defined by Horvat et al. (2020).

transition. For both bare and snowy features, regions smaller than  $100 \text{ m}^2$  have fractal dimension fd around 1.25 while features larger than  $150 \text{ m}^2$  have fd around 1.37. From the spatially-averaged reflectance and irradiance shown in Figure 7, we have arrived at the intuitive result that snowy surfaces are of higher total reflectance than bare surfaces— and that red/near-infrared reflectance is particularly lower for bare ice. Our geometric analysis provides us with the ability to parse this effect by feature size. Figure 10 shows the surface reflectance spectra partitioned by surface type (bare and snowy) and characteristic length scale in bins separated by  $2 \text{ m} (\pm 0.5 \text{ m} \text{ binwidth})$ . The dotted yellow trace indicated by the black asterisk marks the mean spectral reflectance for all features with D > 20 m.

From these spectra, we observe that the spatially-averaged relationships tend to hold at individual scales: for features of a particular size, snowy features will be more reflective than bare features, with bare features especially lower in reflectance at the red/near-infrared wavelengths. However, we also find that the reflectance of snowy ice features tends to vary evenly with size across the visible and near-infrared wavelength range up to  $D \approx 14$  m, at which point the spectral reflectance loses sensitivity to feature size and the spectra for  $D \ge 14$  m in Figure 10b overlap. Bare ice features behave quite differently (Figure 10a).

**Figure 9.** a,b Bare and snowy ice feature size probability density functions (left axes) & normalized cumulative area (right axes). PDF marker color corresponds to the mean feature color for a particular size bin. c,d Bare and snowy feature area-perimeter plots, with individual measurements overlaid with a bivariate histogram. Two lines in panels c & d correspond to fractal dimensions of 1 and 2; the piecewise linear functions correspond to fractal dimensions computed over two regimes: areas ranging from  $2.6 \text{ m}^2 - 100 \text{ m}^2$  and  $150 \text{ m}^2 - 100,000 \text{ m}^2$ .

Figure 10. Spectral reflectance  $\rho_s(\lambda)$  for bare and snowy ice (left and right panels, respectively). The color of each trace indicates the mean effective feature diameter D for which the spectra were bin-averaged. The dotted trace corresponds to the average for D > 20 m. The translucent rainbow bar at the base of the plot indicates the approximate color corresponding to each particular wavelength.

All reflectance spectra show substantial decline in the near-infrared range, as one might expect from the spatially-averaged spectra shown in Figure 7b and by merit of our very definition of a "bare" feature (near-infrared to blue spectral reflectance ratio). However, reflectance does not increase monotonically with bare ice feature size; rather, reflectance peaks for features with  $D \approx 18$  m. Furthermore, the spectral reflectance averaged over all features with D > 20 m is substantially lower in the near-infrared than all other spectra, offering insight into the nature of bare ice radiative signature variation with feature scale.

Our 3-level classification scheme simplifies the complex reality of the sea ice surface. In truth, an individually-identified contiguous feature may vary a great deal in its radiative properties over space. In order to describe this variation while keeping the clarifying simplicity of our feature categorization, we computed the inter-percentile (interquartile and interdecile) sub-feature variation as a function of feature scale (Figure 11). We find that variability in spectrally-averaged reflectance is significantly smaller across all features with than it is across larger features. For larger features ( $D \ge 6$  m), the variation of spectrally-averaged reflectance is much greater for bare ice than for snowy ice.

Figure 11. Variation of bare and snowy (a & b, respectively) ice feature surface reflectance  $\rho_s$  with feature size. Marker color indicates the mean feature color; thick and thin vertical lines mark the interquartile and interdecile ranges of  $\rho_s$  for each bin. Rust-colored diamonds indicate the ratio between the interdecile range and median value of  $\rho_s$ , represented as a percentage. The markers in the gray shaded region represent values computed for all features with D > 20 m.

#### 4 Discussion

The satellite imagery in Figure 1 shows Reviewer #2: that the transition of the ice surface from predominantly snow-covered (white) to predominantly bare or ponded (grey or blue-green) Reviewer #2: that are remarkable degree of melt and surface degradation occurred between 5/6/2019 and 5/9/2019, meaning that our aerial observations were made over ice that was Reviewer #2: already in an advanced state of degradation. The relatively warm, moist air mass that arrived in the hours before 5/8/2019 may have been the principal culprit for this change. Skyllingstad and Polashenski (2018) found that synoptic weather events transporting warm, moist air over sea ice (and driving sensible/latent heat flux) are the key trigger of incipient surface melt, after which changes to surface albedo result in solar radiation being the principal driver of heat uptake by the sea ice surface. Light transmittance and subsurface light absorption is markedly higher for first-year ice than for multi-year ice (Nicolaus et al., 2013). The blue-green appearance of the bare ice may have been a consequence of the thin ice conditions during our field campaign (thickness determined through *in situ* sampling by Witte et al. (2021) and Mahoney et al. (2021)). In these conditions, it is reasonable to expect that light may transmit through the ice Reviewer #1. Reviewer #2: and underlying water, reflecting off the bottom and reducing the observed reflectance across all wavelengths. We expect this effect to be strongest for the largest bare ice features, where lateral scattering effects are minimal. The tendency of bare ice features with D > 18 m to get darker with increasing size (Figures

10a, 11a, & 12a) Reviewer #1. Reviewer #2: may therefore be partially attributable to the thin ice found at this advanced stage of melt. We also suspect that chromophoric dissolved organic matter (CDOM) concentrations were moderate to high in the coastal waters of Kotzebue Sound. However, it is difficult to say whether or not these effects (sediment, ice thickness, CDOM) had bearing on the scale-dependent reflectance behavior observed by our airborne instrumentation. It should be noted that the sea ice in Kotzebue sound was qualitatively determined to be rich in sediment from the Kobuk and Noatak Rivers, which comports with the gray-brown color frequently observed via our hyperspectral imager (Figure 7a). The presence of sediment is understood to greatly enhance the absorption of solar radiation across the visible and near-infrared wavelengths (Perovich, 2017). The absence of multi-year ice, abundance of sediment, and potentially widespread snow ice formations (Mahoney et al., 2021) may all help to explain how the landfast ice progressed from snowy and white to rotten and broken in one week's time.

The radiometric observations performed as part of the present study were parsed along three principal axes: feature type (i.e., ponded ice, snowy ice, and bare ice), feature geometry (i.e., area, perimeter, or some derivative thereof), and wavelength (i.e., in a spectral sense). The simplest form of the results from this work is shown in Figure 7, where the variation of surface reflectance (wavelength-averaged and wavelength-dependent) is displayed as a function of net solar irradiance. The results of Figure 7a are intuitive, and in agreement with the findings of Yackel et al. (2000): regions with a higher fraction of white/snowy ice have higher total surface reflectance and are associated with lower absorption of solar radiation. The surface reflectance spectra shown in Figure 7b offer a view of sea ice that is analogous to the one presented by Grenfell and Perovich (1984), in which measurements of spectral albedo were performed over sea ice with a variety of surface types. For the bare ice and melt pond conditions similar to those observed during our field campaign, there was found to be a steep decline with wavelength for wavelengths between 600-800 nm, with white ice showing a far more gradual decline with wavelength.

Melt ponds draw a great amount of attention in the literature related to sea ice radiative properties, and for good reason. On thick, multi-year ice with persistent snow cover (and therefore generally high albedo across the visible spectrum), melt ponds represent islands of exceptional radiative penetration and under-ice heat absorption. The thin and degraded first-year ice of Kotzebue Sound that served as the subject of our observations in May 2019 held a vanishingly small number of melt ponds: less than 0.5% of the imaged surface area. Although the classification algorithm has been validated via in situ observations by other researchers (König and Oppelt, 2020), we did not perform any ground-truth measurements during our field observational period. It is therefore possible that our approach under-counted melt ponds. This may have impacted our ability to fully characterize melt pond fractal dimension: we did not observe a transition in melt pond fractal dimension from 1 to 2. This transition corresponds to ponds of increasingly serpentine, complex arrangement, and is often observed (Hohenegger et al., 2012) and reproduced via physical modeling (Bowen et al., 2018). Nevertheless, as shown in Figure 8, our observations of the melt pond spatial distribution showed strong agreement with previous observations (Huang et al., 2016; Perovich et al., 2002) and the geometric model of Popović et al. (2018).

The geometric analysis frameworks often applied to melt ponds is in fact well-suited towards the other surface types classified here: snowy & bare ice (Figure 9). For those surface types, we found that small, relatively dark features were most plentiful, with larger and more reflective features somewhat scarce. The behavior in these two feature size clusters appeared to follow two distinct power laws. For snowy features, these are  $P(D) \propto D^{-1.3}$  for 2 m < D < 5 m and  $P(D) \propto D^{-3.5}$  for 10 m

m < D < 20 m. The intermediate domain of 5 m < D < 10 m corresponds to a range in perimeter of 40 m - 80 m for a feature with area 100 m<sup>2</sup>; it is over this span of values that we also observe the transition of fractal dimension from approximately 1.25 to 1.37. This shift indicates that larger, more reflective snowy and bare ice surface features are more complex in shape than smaller, darker features—but only a bit more so. The analogous transition for melt ponds (Bowen et al., 2018; Hohenegger et al., 2012) is quite stark: from fractal dimension just above 1 (very nearly circular ponds) to fractal dimension approaching 2 (very nearly a space-filling curve). This comparison indicates that, while feature complexity of bare and snowy ice surfaces increases with size, it does so only up to a moderate level. This indicates that large features tend to be bulky, with well-defined interior and exterior regions.

Figure 12. a Variation of  $\rho_s$  with feature size D, expressed as percent difference from the area-weighted average surface reflectance and partitioned into bare (teal circles) and snowy (gray squares) categories. Shaded regions indicate interquartile and interdecile ranges of variation. **b** normalized cumulative area of features with diameter less than or equal to D.

305

The reflectance spectra depicted in Figure 10 offer a more detailed view of these surface feature characteristics. The spectra show an increase in snowy ice reflectance with feature sizes up to  $D \approx 14$  m, with larger patches no longer varying with size in spectral reflectance. This is compatible with our previous description of surface features as fairly simple in shape with well-defined interior regions. The observed behavior for bare ice surfaces is quite different, varying non-monotonically with feature size. For bare ice regions, reflectance increases with feature size up to  $D \approx 18$  m, only to decrease with size (especially in the near-infared range). This reduction appears to become substantial for the features with D > 20 m, patches which constitute over 55% of the total observed bare ice by area. As discussed earlier in this section, it may be the case that our classification scheme is underdetecting ponded sea ice. However, the stark difference between snowy and bare reflectance spectra indicate that we are identifying meaningfully different surface types, even if there is some uncertainty regarding the identification of melt ponds. The sub-region variability in reflectance (Figure 11) provides a complementary description of this distinction. For both bare and snowy surfaces, variability in reflectance increases with scale as features begin to develop fringes, tapering off as darker fringes become a smaller portion of the overall feature area. The scale dependence of this effect appears coincident with the power law and frac-

tal dimension transition scales, shoring up the interpretation that small, loosely-connected regions of uniformly-low reflectance are plentiful.

A broader question remains: how important are these size-dependent effects in the context of the radiative balance at the sea ice surface? In order to address this question, we reformulated the information provided in Figure 11 with respect to the mean total surface reflectance: Figure 12a shows the variation of surface reflectance with feature size for both bare and snowy

ice features. For bare ice, both small features and large features appear to be darker than features of moderate size. We are not able to offer an explanation for 18 m being a lengthscale of particular significance. It is likely the case that the phenomena responsible for the relative feature darkness at small and large sizes differ from one another. For example, "small" features are often complex in shape (small area/perimeter ratio), whereas for large features, it may be that lateral scattering is less pronounced.

However, these small features still represent a small fraction of the overall sea ice surface area. In order to provide a broader view of the observed sea ice radiative characteristics, it is important to also consider large patches which are few in number but great in total surface area: large features (D > 20 m) occupied  $\approx 60\%$  of the sea ice surfaces we sampled (Figure 12b). Snowy ice features increased in mean reflectance (up to  $\approx 20\%$  higher reflectance than average) with size up to some critical scale at which the patch interior is sufficiently protected from the darker, more absorptive patch fringes. For bare ice features, however, the largest regions which occupy the bulk of the bare ice surface area are darker:  $\approx 20\%$  lower total reflectance than average, with particularly low reflectance in the near-infrared. Based on the highly variable reflectance observed for the largest bare ice patches (Figure 11a), it may be that these regions include cracks, melt water, suspended sediment, and other characteristics which render them vulnerable to increasing absorption of solar radiation.

These findings indicate that the degree to which sea ice reflects incoming solar radiation is strongly impacted by the spatial characteristics and geometric properties of features which constitute the sea ice surface. We expect that efforts which seek to model changes to Earth's climate will be advanced by incorporation of these effects, and we eagerly anticipate future studies which investigate this scale-dependence in ice-albedo feedback mechanisms. Indigenous communities who rely on the sea ice as part of their way of life may benefit from such improvements to climate forecasting capabilities. Furthermore, quantification of solar radiation uptake that is dependent upon sea ice color and feature size may assist in the determination of structural safety for use in travel and hunting activities.

#### 5 Conclusions

We performed a series of aerial observations of sea ice radiative properties at an advanced stage of melt and breakup. These observations were performed in Kotzebue Sound, Alaska, and occurred in the context of knowledge co-production with with Elders from Kotzebue (study co-authors J.G., C.H., R.J.S., and R.S., Sr.). This collaboration began with the initial project conception, continuing through execution of observations, and extending into analysis. As an example of this knowledge co-production, this study's Elder co-authors noted that when strong winds blow dust from exposed mud bars onto the sea ice, that "dirty ice" melts far more rapidly than the neighboring white ice. This experience highlights the need for high spatial resolution sampling of radiative characteristics of sea ice during the melt process. Our observations allowed us to quantify the mean solar radiative flux into the sea ice as a function of both ice color and spectral surface reflectance. Using high-resolution hyperspectral radiometric imaging, we were able to reconstruct maps of spectral reflectance over the visible to near-infrared range at spatial scales down to 50 cm. This dataset was used to classify the surface into regions of melt ponds, bare ice, and snowy ice and to perform geometric analyses on those regions across the resolved spectrum. Melt ponds were scarce during

our operating period, though the ponds we did observe presented with a scale distribution that agreed with that of previous observations and modeling. We found that \*Reviewer \$2: the reflectance of sea ice features varies with feature size: snow patches get darker patches of snowy ice become less reflective as they get smaller \*Reviewer \$2:, while bare patches get darker as they get larger. The reflectance of bare ice patches varied non-monotonically with feature size, though features larger than 18 m (characteristic diameter) were were observed to become less reflective with increasing size. \*Reviewer #2: This is an albedo feedback that is These are albedo feedbacks which are dependent upon the size of the features which constitute the sea ice surface.

These observations of surface reflectance variation offer insight into the role that feature size distribution and geometry play in the overall radiative balance of sea ice. Diminished surface reflectance leads to enhanced uptake of solar radiation, which in turn hastens the degradation (and ultimately, breakup) of the sea ice. We expect that the size and geometry dependence of surface reflectance works to strengthen positive feedbacks associated with radiation uptake. In summary: for bare ice features within landfast ice, their absorption of radiation is expected to advance the melt of adjacent snow, accelerating the darkening of the feature (and growing it at the expense of the surrounding white/snowy ice). Conversely, large snowy features absorb more solar radiation as they are subdivided and split by melt and degradation, with a greater fraction of their area in contact with darker, serpentine fringe regions. We recognize that climate models cannot afford to directly resolve individual features on the sea ice. However, we hope that the effects described in this paper can be parameterized statistically, incorporating scale-aware feedbacks into models without the need for directly resolving individual features.

In closing, these observations provide a quantitative description of the radiative and geometric properties of features on the surface of first-year sea ice undergoing the late stages of melt and breakup. While snow cover on sea ice is known to inhibit ice-bottom growth in winter by insulating it from the cold atmosphere (Mahoney et al., 2021), the high albedo of snow helps to slow ice melt in spring. Our results suggest that the spatial characteristics of the snow cover may offer predictive skill in determining ice breakup patterns in regions where ice melt is dominated by the radiative flux balance. Although the observational period was short, the high spatial and spectral resolutions of the sensing approach deployed here have yielded a rich characterization of the sea ice surface. We anticipate that the results of this work will be used to inform future, larger-scale field observational and modeling efforts aimed at providing a comprehensive description of the sea ice surface properties.

# Appendix A

The TSPI solution produced through post-processing of data obtained from the Base payload's INU provided the aircraft position and attitude at 50 Hz. This was interpolated onto the time vector of the radiometric instrumentation of the VNIR payload. Three attitude angles—pitch, roll, and yaw—are shown in Figure A1a; note that the yaw angle represents variation about the mean heading. An RGB representation of 60 seconds' worth of calibrated (but not georectified) surface reflectance data are shown in Figure A1b. Vectors originating from the camera focal point and terminating at each point in this array were initialized; the position corresponding to each spatial measurement of surface reflectance was determined by rotating each vector by the instantaneous UAV attitude in three dimensions and finding its intersection with the ellipsoid defining the mean ocean surface. The output of this process is shown in Figure A1c. Another illustration of this VNIR pushbroom orthrectification is provided by Zappa et al. (2020).

**Figure A1. a** Time series of aircraft pitch, roll, and yaw angles; **b** color image generated from 40-second snippet of imager's red, green, and blue channels; **c** corresponding orthorectified mosaic.

**Figure A2.** Color mosaics showing the same patch of sea ice from flights on 5/10/2019 **a** and 5/14/2019 **b**. Panels **c** and **d** provide the corresponding spatial characterizations of surface ice type based on NIR:B and the melt pond detection algorithm of König and Oppelt (2020).

Code and data availability. Historical meteorological data from the Kotzebue airport were obtained from the National Centers for Environmental Information at https://www.ncdc.noaa.gov/cdo-web/datasets/LCD/stations/WBAN:26616/detail. The terrestrial and UAV-obtained datasets (as well as all computer codes used to process them to produce the figures shown in this document) are available through the Columbia Academic Commons (DOI: 10.7916/rrbv-k026).

Author contributions. NJML planned the UAV flights (with CJZ), performed the UAV data processing, and organized the analysis. CJZ directed the UAV operations and conceived of the overall project with ARW, AS, DDWH, SB, and AW. ARW contributed expertise related to sea ice geophysics during the manuscript writing process. JG, CH, RES, and RS provided in situ expertise during field operations and throughout analysis and writing. CRW, JML, and KET collected land and sea-ice based observations in support of the effort. NJML designed and wrote the manuscript with feedback from all authors.

Competing interests. The authors declare that they have no competing financial interests.

*Disclaimer.* Publisher's note: Copernicus Publications remains neutral with regard to jurisdictional claims in published maps and institutional affiliations.

Acknowledgements. This work was supported by a grant from the Gordon and Betty Moore Foundation. We are grateful to the Qikiqtagrunmiut people and community of Kotzebue, Alaska. The sharing of their Indigenous Knowledge was an integral part of the design
and execution of this work. We thank the Native Village of Kotzebue and the University of Alaska Fairbanks (Chukchi campus) who provided meeting facilities and the US Fish and Wildlife Selawik Refuge, who provided crucial logistical support. We thank Scott Brown, Tej
Dhakal, and Ryan Harris for their engineering expertise in the development and field operation of the scientific payloads over a period of
many years. We thank Cory Rosene, Scott Bowers, and Sean Culbertson for their work in piloting and maintaining the aircraft during our
field operations in Alaska.

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
