# Peer review of "The Radiative and Geometric Properties of Melting First-Year Landfast Sea Ice in the Arctic"

_EGUsphere, 2023_

## Author Response (AR1)

**REVIEWER #1**

*This manuscript presented a study on the radiative and geometric properties of sea ice observed by a fixed-wing UAV, which is interesting to Arctic sea ice society. The geometric properties part (Figures 9, 10) was something common study, which didn't present novel information. The second part, the dependence of reflectance on ice size is indeed novel and could be useful as thinking about the impact of incorporating ice size into new sea ice models. However, I am confused by the motivation of this work. The problem that this manuscript seeks to solve needs to be clarified.*

We are grateful to the reviewer for their feedback. After reading the feedback from both reviewers (and the editor), we determined that we needed to more effectively communicate the fact that observations were made over landfast sea ice- and that all of the feature spatial/geometric analysis was performed with respect to regions in/on that landfast ice, not floes within a marginal ice zone. Text has been added to section 2.1 to this effect.

We have responded to each of the reviewer's concerns on a point-by-point basis:

***Major concerns:***

*Figures 11, 12, 13: It is reasonable that ice reflectance increases with the floe size, because of the lateral loss of light out the ice boundary. Furthermore, when the size is big enough (e.g. > 15 m), the reflectance is nearly identical. This result agrees with other simulated results. However, I don't understand why the reflectance of ice with size was 20 and > 20 became smaller.*

This was identified by both reviewers, and we believe it is an important point for discussion. As shown in our Figure 13, for bare ice, both small features and large features appear to be darker than features of moderate size. We do not have an explanation for 18 m being a lengthscale of particular significance. It is likely the case that the phenomena responsible for the relative feature darkness at small and large sizes differ from one another. For example, "small" features are often complex in shape (small area/perimeter ratio), whereas for large features, it may be that lateral scattering is less pronounced. This content has been added to the manuscript's Discussion section.

*It seems that the manuscript didn't present the observing time of UAV flights. The figure shows ice reflectance but not albedo. So, there is a question, whether the effects of solar zenith angle on ice bidirectional reflectance been considered?*

The flight times have been provided in section 2.1 of the manuscript along with the associated solar zenith angles.

*I also didn't see the view angle of the radiance sensor. At a flight altitude of 1000 m, its field of view may be over 40 m (assume the view angle is 5°). So, it is doubly that if the observed reflectance is from bare ice or ice and seawater. If it is the latter one, the word*

*'size' used here is misleading. The changing radiance is mostly due to the changing ice concentration but not size. Then, the novelty of this work disappeared.*

The imaging radiance sensor view angle (both full swath and pixel IFOV) is provided in Table 1. As mentioned in the surrounding text, the ground sample distance is approximately 50 cm at a flight altitude of 1000 m.

Regarding the possibility of including seawater in our observations- open water is not included in analysis, with a rejection criterion specified at the beginning of section 2.2.

***Other concerns:***

*The abstract need re-write, and there are lots of introduction but less results.*

We agree that the abstract was heavily laden with introductory material. We trimmed three lines from the abstract and rephrased introductory sentences. Now, half of the text in the abstract is devoted to summarizing results.

*There are too many abbreviations in the manuscript that weren't defined in the main text. So, it is difficult to understand the figures.*

VNIR and RAD are explicitly defined at the beginning of section 2.1. Additionally, mentions of GPS (US-based constellation) have been replaced with GNSS (the general term), with the latter acronym defined in section 2.1 in the vicinity of L95.

*L36-39: The Arctic amplification can't be regarded as a total result of the changing ice situation.*

We have re-worded this sentence, stating that the changing ice situation "has contributed to..." Arctic amplification.

*Figure 3: It seems this figure didn't show any useful information.*

We agree that the old Figure 3 was minimally useful. We have removed it in order to improve the flow of the manuscript.

*Section 3.1 (Figure 8): I don't understand what this figure seeks to present. The difference in $\Delta E_s$ was mostly controlled by the reflected radiance. It is a matter of course that mean or spectral reflectance decreases with increasing $\Delta E_s$.*

The left panel provides reader with a visual depiction of the sea ice color and shows the range of conditions (in reflectance and delta E_s) observed. Right panel provides spectral reflectance binned by net shortwave irradiance; reader can connect level and shape of spectra with marker color/darkness in panel A. As pointed out by the reviewer, the relationships between these quantities are a matter of course. We included this figure not for the purpose of presenting novel results, but to provide background and context. Also- we note that the ordinate of panel (a) is properly labeled (rho_s, sr^-1), but the ordinate of panel (b) has been noted in the caption to be of units sr^-1 nm^-1.

Review of "The Radiative and Geometric Properties of Melting First-Year Sea Ice" This manuscript presents observational data from a field campaign deploying UAV with payloads designed to characterize the radiative properties of the sea ice surface. Data were collected over landfast Arctic sea ice during the melt season as the ice transitioned from snow covered to bare and melting. During this time the floe size distribution was rapidly evolving from larger floes to smaller floes.

The text is written very clearly and concisely. The figures are clear and informative. There is a lot of information here and the results are of interest to the sea ice radiative transfer modeling community.

We are grateful to the reviewer for their feedback. We have responded to each of the reviewer's concerns on a point-by-point basis:

I find the conclusions intriguing, but would be informative to have more explanation of the effects that were documented. For example, I can think of a variety of explanations for the results showing sensitivity of reflectivity to feature size:

- Smaller features are more influenced by their perimeter, perimeters are subject to some strong effects that can alter their reflectivity, such as lateral melting, wave wash.
- Light leakage: do photons propagate laterally out of floe boundary before they can be backscattered? Likewise, do photons enter the ice from the side and increase the albedo?

It would be helpful to the reader to consider these mechanisms (and there may be more?) and either substantiate them or refute them. I think it might not be that the larger floes are darker, but rather that the smaller floes get light leaked in from the sides?

We needed to more effectively communicate the fact that observations were made over landfast sea ice- and that all of the feature spatial/geometric analysis was performed with respect to regions in/on that landfast ice, not floes within a marginal ice zone. Text has been added to section 2.1 to this effect.

Am I interpreting correctly that 18 m is the size scale that roughly divides whether a floe is on the "large" or "small" size? From a radiative transport perspective, that "boundary" sounds very large to me.

This was identified by both reviewers, and we believe it is an important point for discussion. As shown in our Figure 13, for bare ice, both small features and large features appear to be darker than features of moderate size. We do not have an explanation for 18 m being a lengthscale of particular significance. It is likely the case that the phenomena responsible for the relative feature darkness at small and large sizes differ from one another. For example, "small" features are often complex in shape (small area/perimeter ratio), whereas for large features, it may be that lateral scattering

is less pronounced. This content has been added to the manuscript's Discussion section.

I am a bit surprised that dust and sediment were mentioned on a couple of occasions, but not explicitly considered in the explanation. Further, not much was said about the green appearance of the ice cover. Is this a result of biologic activity? Or dust/sediment?

At the beginning of the Discussion section, we posit that the presence of sediment may have been partially responsible for the rapid degradation of the sea ice in Kotzebue Sound during our study period. We have added text to the Discussion which ties the eyewitness observation to the mean sea ice color shown in (now) Figure 7a.

Regarding the blue/green appearance of the bare ice: this may be a consequence of the thin ice present during our field operations (Witte et al., 2021; Mahoney et al., 2021). In these conditions, light may transmit through the ice and reflect off the bottom. We also suspect that CDOM concentrations were moderate to high in the coastal waters of Kotzebue Sound. However, it is difficult to say whether or not these effects (sediment, ice thickness, CDOM) had bearing on the scale-dependent reflectance behavior observed by our airborne instrumentation.

There are a lot of interesting results and intriguing discussions in this manuscript. I feel that the overall story of how the pieces fit together could be tightened up.

We are grateful for the reviewer's suggestions for improving the manuscript. We feel that the expanded discussion points mentioned here will help to tie some of the disparate components together.

**Minor comments:**

Title: add words "Arctic" and "landfast"

This has been done.

Line 3: matter of weeks? Sure, some areas undergo the transition from snow covered sea ice to open water in a matter of weeks, but that process takes a lot longer in other regions, depends on location and ice type.

We have edited the abstract following the suggestions of another reviewer. The line in question has been removed.

14-15: "than average" than average of what? All features?

We have clarified this within the text. As stated in the caption of (now) Figure 12, it is the area-weighted average surface reflectance observed across all observational cases.

40: "Arctic system" less resilient to change? Of should this say "Arctic ice cover" is less resilient?

We have re-worded this to refer to "Arctic ice cover"

Fig 1: this figure shows the color of the ice, but why choose the same color for the mask?

We have changed the land color to a sandy (light brown) color.

160: estimation, retrieval of feature length scale hasn't been sufficiently motivated, why should it be measured? Section 2.2 is 'how', but there is no 'why'

We recognize that this component of the study has not been adequately justified in the present text. In the introduction (the end of section 1.1), we refer to Popovic et al. [2019] and Horvat et al. [2020] in order to motivate the scale and geometry-based analysis in the present manuscript.

Fig 11: "thickness of each trace corresponds to the mean effective feature diameter…" not clear how "thickness" is being used here. all look same thickness (width?) to me. Or does 'thickness' refer to "D" and hence the gray/yellow shade?

That legend corresponds to a previous version of figure; we have updated the legend to properly match the figure.

329-331: "…to strengthen positive feedbacks associated with radiation uptake. In short: the tendency of large blue-green features to absorb radiation increases with their size, while large snowy features absorb more solar radiation as they are subdivided and split by melt and degradation." I don't understand the mechanism whereby an increase in absorption with increasing size is a positive albedo feedback. That would suggest that as the floe size decreased (due to melt, increased absorption), the absorption would also decrease (which sounds like a negative feedback)

We have rephrased this text in order to more effectively communicate our description of the feedback effect.

---

## Author Response (AR2)

***Reviewer #1:***
*The authors have made revisions to the manuscript that have addressed some minor concerns. However, the major concerns still exist.*

*Major concern 1:*
*The author still has not provided a clear explanation for the result shown in Figure 10, 11, 12. Only two lines text were newly added. As stated in the revised text, the author guessed that the lateral scattering of large features is less significant. However, the related model results suggest that scattering from the lateral ice boundary can be ignored when the ice size is larger than 10 m. So, this cannot explain why the reflectance of ice with size of 20 m smaller than size of 18 m.*

We agree with the reviewer that a satisfactory explanation for those results was not provided in the previous manuscript draft. After consideration of the feedback provided by both reviewers, we find the most reasonable explanation for this observed behavior to be one related to the thinness of the sea ice. In short: the sea ice was quite thin at the time of our observations, so it is justifiable to assume that there was a sensitivity of surface reflectance to ice thickness. This sensitivity is expected to be greatest for the large patches of bare ice, reducing the surface reflectance for those regions. Text to this effect has been added to the Discussion (L235-238).

*Major concern 2:*
*The flight times have been added in the main text. There is a difference of nearly 5 hours between the flights. The author stated that great care was taken to ensure that ambient radiative conditions did not differ too greatly from one flight to the next. Let's just assume that the radiative conditions are the same at noon and nightfall. However, the solar zenith angle inevitable changed by about 20°. (Note: the angles added in Section 2.1 were elevation angles, not zenith angles). For ice with an albedo of 0.5, a 20° change in zenith angle can result in a 20-25% change in ice albedo. However, this effect was not taken into account in the whole manuscript, making most of the results doubtful. The authors arbitrarily attribute all changes to size effects, which cannot explain many of the results.*

We have corrected our erroneous description of the angles in Section 2.1 as zenith angles (they are now properly referred to as elevation angles). We

note the reviewer's concern regarding the variation in flight times—and the potential for introduction of a confounding variable. The greatest difference in solar elevation angle from one flight to any other occurred between the time of the final RAD flight (on 5/15/2019) and the other two on 5/10/2019 and 5/14/2019. The RAD and VNIR flight paths did not overlap on our final day of operations due to operational constraints. As a result, data from that day of flight were not included in our analysis- a fact which is now communicated in the manuscript text at the end of Section 2.1. With respect to the four flights remaining under consideration, the differences in solar elevation angle were 0.8 degrees for the RAD payload and 5.13 degrees for the VNIR payload, limiting the degree to which our observations depended on solar angle.

**Reviewer #2:**

*It seems likely that the sea ice in this study was thin enough for the albedo (reflectance) to show sensitivity to bare ice thickness. Is it possible that bare areas (of size > 18 m) get darker as they get larger because of this thickness sensitivity?*

We are grateful to the reviewer for offering this potential explanation for that observed behavior. We find it to be the most reasonable interpretation of the available facts. L235-238 now contain text which offers this account of the observations, connecting the discussion text of the preceding lines with the relevant figures.

*Line 163-164: would be helpful to indicate symbols used on plot axes in the text and in the caption: "...with mean visible and near-infrared surface reflectance (ρs) plotted against net solar irradiance (deltaE_s).*

The text as originally written gave the false impression that two distinct quantities (visible AND near-infrared surface reflectances) were being plotted alongside one another. The plot in fact shows the surface reflectance averaged over the visible band (as previously mentioned in what is now L96-97 of the text). The text (and caption of Figure 7) have been edited to improve communication of this important detail.

*Fig 7a: color scale? Also, not clear, what is distinction on plot between visible and near-infrared surface reflectances?*

The caption of Figure 7 indicates that the color of the markers in panel a are representative of the mean sea ice color of the features within that bin. This is meant to provide a qualitative visual aid to the reader. The caption has been edited to clarify that this is visible light surface reflectance.

*Fig 9(a, b): color scale?*

Please see response to previous comment.

*223 - 225: Not a sentence. Rewrite. Or "...shows that for the transition..."?*

This chunk of text has been edited to ensure that it is both grammatically correct and easy to understand.

*344: Would it be more accurate to say that "..while bare patches of size greater than 18 m get darker as they get larger."?*

We agree with the reviewer that the description given in (former) L344 is too simplistic. The non-monotonic behavior of mean reflectance with bare feature size does complicate this picture. We have modified the text to more accurately describe observed dependence of bare ice features on characteristic size.